# Evaluation of Antifungal Activity of Ag Nanoparticles Synthetized by Green Chemistry against *Fusarium solani* and *Rhizopus stolonifera*

**DOI:** 10.3390/nano13030548

**Published:** 2023-01-29

**Authors:** J. M. Moreno-Vargas, L. M. Echeverry-Cardona, L. E. Moreno-Montoya, E. Restrepo-Parra

**Affiliations:** Laboratorio de Física del Plasma, Universidad Nacional de Colombia, Sede Manizales, Manizales 170001, Colombia

**Keywords:** Ag nanoparticles, green chemistry, antifungal, phytopathogens, avocado

## Abstract

Silver nanoparticles (AgNPs) have aroused great interest for applications as fungicides in agriculture. This study reports the synthesis of AgNPs by green chemistry using silver nitrate (AgNO_3_) as the precursor agent and a coriander leaf extract as the reducing agent and surfactant. The evaluation of their antifungal properties was carried out when placed in contact with *Fusarium solani* and *Rhizopus stolonifer* phytopathogens. The extract and AgNP characterizations were performed using UV–Vis spectroscopy, X-ray diffraction (XRD), Fourier Transform Infrared Spectroscopy (FTIR), dynamic light scattering (DLS) and scanning electron microscopy (SEM). The evaluation of antifungal properties was carried out by exposing the phytopathogens to different concentrations of AgNPs in PDA (Potato Dextrose Agar). It was found that it was possible to identify the presence of flavones and flavonoids in the extract, compounds that were also involved in the synthesis process of AgNPs. In addition, the UV–Vis analysis of the obtained AgNPs by green chemistry showed resonance peaks at around 428 nm. Furthermore, a high distribution of AgNP sizes, with high concentrations of below 100 nm, was identified, according to DLS measurements. Using SEM images, the information provided by DLS was confirmed, and a crystallite size of 29.24 nm was determined with the help of XRD measurements. Finally, when exposing the phytopathogens to the action of AgNPs, it was concluded that, at a concentration of 1 mg/mL AgNPs, their growth was totally inhibited.

## 1. Introduction

Nanotechnology has attracted great interest and is projected as a promising solution for various problems that afflict humanity in various fields such as medicine [1,2], the food industry [3,4], power generation [5], environmental protection [6,7] and agriculture. The aforementioned applications are due to the fact that nanometric-scale materials exhibit exceptional chemical, optical, and electrical properties with respect to homologous materials of larger dimensions [8]. In addition, nanomaterials of metallic origin, in particular nanoparticles, demonstrate special behavior when interacting with agents in their environment [9] that generates the production of reactive oxygen species that intervene in cytotoxic processes, which has enabled the application of these nanomaterials as antibacterial and antifungal agents [1,2,3,4,10,11,12,13].

Recently, copper, gold and silver nanoparticles have become important in the agroindustry sector for the treatment of diseases and conditions generated by phytopathogens that attack crops; they constitute a valid alternative to replace traditional fungicides whose toxic effect on the environment and humans is considerable [6,14,15,16]. In this sense, investigative efforts have been concentrated on the development of new alternatives to combat the pathological effects on crops of different products caused by different types of fungi. Boxi et al. [17] synthesized hollow TiO_2_ nanoparticles doped with Ag nanoparticles through a chemical route. Microbiological tests were carried out with the obtained nanoparticles, exposing fungal cultures of *Fusarium Solani* and *Venturia inaequalis* to these nanoparticles. The authors evaluated the response of the phytopathogens under conditions of darkness and the irradiation of visible light, demonstrating better antifungal response in the presence of visible light. 

Copper nanoparticles have also been considered as an alternative for the treatment of phytopathogens that attack different crops. Pariona et al. [18] synthesized copper nanoparticles by a chemical route. With the obtained nanoparticles, in vitro and in vivo microbiological tests were carried out to evaluate the antifungal response when exposing phytopathogens, *Fusarium*, *Verticillium* sp. and *Neofusicoccum* sp., to the action of the nanoparticles. It was determined that these nanoparticles have promising antifungal properties, and different levels of inhibition were observed depending on the exposed fungus. Malandrakis et al. [19] evaluated the synergistic fungicidal action between silver nanoparticles and a conventional chemical fungicide (thiophanate methyl or fluazinam, both in vitro). It was concluded that the fungicidal action of the mixture between the nanoparticles and the traditional fungicide was greater than the action of each of these elements separately; in the same way, better action was observed in the inhibition of the growth of the phytopathogen by individually applied Ag nanoparticles than that shown by the traditional fungicide. Le et al. [20] also synthesized silver nanoparticles, evaluated their antifungal behavior and determined that the silver nanoparticles are promising fungicides for various applications of medical and agricultural fields. 

The nanoparticles presented in this work were obtained through green chemistry, replacing reducing agents and chemical surfactants with plant extracts, which allowed us to reduce costs and the environmental impacts generated by the chemical reagents. In this report, two types of silver nanoparticles were synthesized, one using an Achyranthes aspera extract and the other using a Scoparia dulce extract. Their antifungal efficacy was assessed via interactions with three types of phytopathogens, *Aspergillus niger*, *Aspergillus flavus* and *Fusarium oxysporum*. It was determined that depending on the extract used in the synthesis process, the morphological properties of the nanoparticles changed; additionally, we observed a positive effect on the antifungal behavior of the silver nanoparticles, which did not generate complete inhibition at the concentrations used but did allow us to infer that greater inhibition could be achieved in the growth of phytopathogens at higher concentrations.

The purpose of this study was to evaluate the effect on the growth of the fungi *Fusarium solani* and *Rhizopus stolonifer*, which attack the avocado plant (*Persea americana*), by placing them in contact with AgNPs obtained by green synthesis using coriander plant leaf extracts as the reducer and surfactant (*Coriandrum sativum*). In order to identify the functional groups present in the extract of coriander leaves, Fourier Transform Infrared Spectroscopy (FTIR) was used, and in order to understand the structural and morphological characteristics of the AgNPs, X-ray diffraction (XRD), scanning electron microscopy (SEM) and dynamic light scattering (DLS) were used. The evaluation of the response of the phytopathogens *Fusarium solani* and *Rhizopus stolonifer* when exposed to silver NPs was conducted through an experiment with PDA (Potato Dextrose Agar) in Petri dishes, subjecting the fungi to concentrations of 0.1 mg/mL, 0.2 mg/mL, 0.5 mg/mL, and 1 mg/mL.

## 2. Materials and Methods

### 2.1. Preparation of the Precursor Agent 

In a beaker with 30 mL of deionized water placed on a magnetic stirrer plate at 600 rpm and a temperature of 60 °C, silver nitrate particles (Panreac, Barcelona, CAT, ES) (AgNO_3_) were dissolved until we obtained an aqueous solution with a concentration of 0.01 molar (0.01 M). These values for the experiment were chosen according to the literature [21].

### 2.2. Preparation of the Reducing Agent

The main activities carried out to obtain the extract of coriander leaves are detailed below and shown in Figure 1.
We obtained the coriander leaves and manually removed the roots and stems.We weighed the leaves and removed impurities. In a 500 mL beaker, we placed 10 g of leaves and washed them with distilled water to remove sand, dust or organic residues.We dried the leaves. These were placed in an electric resistance oven (200 W) at a temperature of 40 °C for 24 h.We reduced their size. In a maceration mortar, the leaves were reduced to smaller pieces and then mixed with 100 mL of distilled water.We obtained the extract. The mixture was subjected to a temperature of 100 °C for 20 min with constant magnetic stirring at 600 rpm.We filtered the initially obtained extract in order to eliminate solid residues.Finally, we stored the extract in a plastic container and placed it in an ARCTIKO refrigerator at a temperature of 4 °C for later use.

### 2.3. Synthesis of Silver Nanoparticles 

Once the precursor agent and the reducing agent were prepared, they were mixed. We dropwise added 6.6 mL of pure extract to the aqueous solution, and magnetic stirring was maintained at 600 rpm for one hour at a temperature of 60 °C. After this time and the observation of a yellow–brown coloration (see Figure 2) as evidence of the synthesis of silver NPs, similar to that reported by [22,23,24], this mixture was packed in 15 mL Falcon tubes. Subsequently, the mixture was subjected to centrifugation (INMEDIAT Model 800D) for 15 min at 4000 rpm. As a result of the previous step, the nanoparticles precipitated to the bottom of the container, which allowed the supernatant to be removed and the nanoparticles to be resuspended in 10 mL of deionized water, a process that was repeated twice more in order to clean the nanoparticles. The entire process described above is shown in the precedence diagram in Figure 3.

### 2.4. Techniques of Characterization 

#### 2.4.1. Characterization of the Extract

In order to determine the functional groups present in the coriander leaf extract and recognize the components that make it a reducing agent and surfactant in the AgNP synthesis process, a drop-shaped sample was placed on a Fourier Transform infrared spectroscope cell Bruker ALPHA II (Bruker, Billerica, MA, USA, EE.UU). This equipment has a spectral resolution of 4 cm^−1^ and a measurement range between 400 and 4.000 cm^−1^. The spectra were automatically analyzed by the OPUS software (OPUS from Bruker, Billerica, MA, USA, EE.UU).

#### 2.4.2. Characterization of the AgNPs

Suspension samples were loaded into a UV–Vis spectrophotometer and subjected to radiation in the spectral range of 200 to 1100 nm in order to obtain absorbance spectra. To determine the hydrodynamic size of the particles suspended in the deionized water, the synthesized samples were subjected to dynamic light scattering (DLS). This process was carried out with Malvern Zsizer equipment (Malvern., Almelo, Overijssel, NLD), which a measuring range between 0.3 nm and 10 µm. Additionally, SEM (Tescan, Brno, Moravia, CZE) measurements were performed on the obtained samples in order to determine the size of the nanoparticles and contrast it with the results obtained by DLS. These measurements were conducted with a Tescan Vega 3 LMU scanning electron microscope (Tescan, Brno, Moravia, CZE) at a magnification range of 2.5× to 1,000,000× and a voltage of 30 kV. X-ray diffraction (XRD) (Bruker, Billerica, MA, USA, EE.UU) tests were performed on the obtained NPs in order to determine their structural characteristics. For this purpose, we used a Bruker Advance diffractometer with a measurement range of 20 to 80 degrees in 2θ, with Bragg–Brentano geometry. To carry out these measurements, the suspension of nanoparticles was deposited and dried drop by drop on a silicon plate until the film had a good thickness. Finally, in order to determine the concentration of the AgNPs present in the suspensions resulting from the synthesis process, some of the suspension was separated and mixed with deionized water to reach 15 mL so that it could later be subjected to atomic absorption from an iCE 3000 series AA spectrometer from Thermo Fisher Scientific (Waltham, MA, USA, EE. UU). The data obtained from the AAS were used to control the concentrations of nanoparticles that were used in microbiological tests with phytopathogens. 

#### 2.4.3. Microbiological Tests

To evaluate the response of the phytopathogens *Fusarium solani* and *Rhizopus stolonifer* when exposed to the synthesized silver NPs, experiments were carried out in triplicate with PDA (Potato Dextrose Agar) in Petri dishes, varying the concentration of the NPs. We worked with a standard sample without treatment (control sample) and four samples for each phytopathogen doped with nanoparticles at concentrations of 0.1 mg/mL, 0.2 mg/mL, 0.5 mg/mL, and 1 mg/mL for a total of 8 treatments evaluated at time intervals of 24, 48 and 120 h.

For the preparation of the standard sample, we used Petri dishes that were previously disinfected and sterilized. In the center of the agar, we created a 6 mm diameter well in which the fungi were inoculated. A similar procedure was carried out to prepare the samples with the silver nanoparticles. These Petri dishes were used with the PDA mixed with 1 mL of the solution of AgNPs in the different concentrations; later, the well was generated in the center of the agar, and 20 µL of the analyzed fungi were planted there. Figure 4 presents a schematic diagram of the experimental design for the evaluation of the effect of AgNPs on the growth of phytopathogens.

## 3. Results

This section presents the results of applying the different characterization techniques to the products obtained in the AgNP synthesis process in order to determine some of their physical, morphological, and compositional characteristics. Likewise, the extract of coriander leaves was characterized in order to understand aspects of its composition. Furthermore, the antifungal behavior of the AgNPs was also evaluated.

### 3.1. Identification of Functional Groups in the Extract of Coriander Leaves

Figure 5 shows the FTIR spectrum of the coriander extract (*Coriandrum sativum*). In the spectrum, the broad band was observed in the wavenumber range from 3700 to 3000 cm^−1^ and centered around 3300 cm^−1^, and it can mainly be attributed to the O-H stretching vibrations of coriander proteins and phenols [23,25,26]. A pair of peaks was observed at wave numbers of 2975 cm^−1^ and 2883 cm^−1^, corresponding to the stretching vibrations of the CH_2_ group [27,28,29]. We also identified a peak at wave number 1654 cm^−1^, corresponding to stretching vibrations of the C=C group; we further identified peaks at 1044 cm^−1^ and 1083 cm^−1^ associated with the stretching of the C–O–C or C–O groups that, together with C=C groups, are related to phytochemicals such as flavones, alkaloids, phenols and anthracenes that can help generate metallic nanoparticles according to the literature [30]. We observed a small fringe at around 1400 cm^−1^, which had a strong relation to the carboxylic acid (COOH) groups originating from carboxyl twisting and bond-stretching vibrations [22]. Finally, a peak was observed at 879 cm^−1^ corresponding to stretching of the C-H bond typical of essential oils [31].

The FTIR spectroscopy results were used to perform the identification of the main functional groups present in the coriander extract. We identified carbonyl, hydroxyl, amide, carboxyl, and phenol groups, which, according to Singh et al. (2018) [30], are capable of reducing metal salts to transform them into nanoparticles.

### 3.2. Absorbance Analysis by UV–Vis

Figure 6 shows the surface plasmon resonance of the absorption spectrum of the AgNPs in water. This spectrum is evidence that the measured nanoparticles produced a strong absorption band caused by the surface plasmon in the visible region centered at wavelengths between 420 and 440 nm, this is a typical range found in the literature for Ag nanoparticles suspended in deionized water with diameters of between 40 and 60 nm. The diameters reported in the literature for these wavelengths was supported by other characterization techniques such as SEM, and was considered that the suspension medium could intervene in the characteristic absorption band of AgNPs according to Mie theory [23,32,33].

The geometry and size of the NPs altered the characteristics of the surface plasmon band and the associated peak in the UV–Vis spectrum, generating a peak shift to longer wavelengths with increasing NP size. This phenomenon was mainly due to the fact that these morphological characteristics modified the electron density of the NPs; as mentioned above, this parameter directly intervenes in the energetic conditions of the vibrating electrons that give rise to surface plasmons [23,33,34]

Taking the above information into account, some authors have focused on approximating the relationship between particle size and peak shift in the UV–Vis spectrum. Accordingly, it can be concluded that the NPs in Figure 6, whose most intense point was at a wavelength of 428.8 ± 1.5, showed the highest concentration of sizes in diameters between 40 and 60 nm [23,34].

However, a considerable width in the absorption band can also be observed in Figure 6, which allows us to infer a wide distribution in the size of the NPs, i.e., little uniformity in the diameter of the NPs, which could be associated with the obtainment technique used in the synthesis process and is consistent with the results obtained using other characterization techniques applied to these NPs [34,35].

### 3.3. Structural Analysis by X-ray Diffraction (XRD)

In Figure 7, a diffraction pattern of AgNPs is presented. In this diffractogram, peaks for 2*θ* values can be observed at 38.2° and 44.4°, which correspond to the (111) plane with a high preferential orientation and the (200) plane, in addition a peak at 33.1° and a small bump at 46.6°, coinciding with silver and silver oxide NPs, respectively. The oxide may have been derived from the oxidation of the metal in the synthesis process, which was developed in an aqueous medium at temperatures above room temperature [36,37,38]. The identification of silver oxide at low intensities in X-ray diffraction is indicative of the presence of a small number of NPs with a surface layer of silver oxide in the form of a core–shell system, such as that reported by Han et al. [39]. 

Using the Debye–Scherer equation (Equation (1)) was calculated and averaged the crystallite size of the peaks corresponding to the AgNPs with the purpose to determine the crystalline nature of these nanoparticles.
(1)$$D=\frac{k\lambda }{\beta \cos\left(\theta \right)}$$

In this equation, values were assigned to the variables as follows: for Scherrer’s constant (*k*), a value of 0.9 was assigned (as recommended by the literature for this type of material); the X-ray wavelength (*λ*) corresponded to 1.54 Ǻ, coherent with the diffractometer radiation source; the mean width of the FWHM peak in radians (*β*) was calculated using a Voigt fit; and the Bragg angle was calculated in radians (*θ*) [40]. As a result, an average crystallite size of 29.24 nm was obtained, consistent with the results found with UV–Vis, allowing us to conclude that the synthesized AgNPs had a high mono-crystallinity.

We used the diffractogram presented in Figure 7 to determine the crystalline structure type and the lattice parameter of the presented material. Peaks located at the 2*θ* positions of 33.1° and 46.6° corresponded to (202) and (132) crystallographic planes, respectively, characteristic of AgO with a cubic structure; these results were consistent with the reference literature (JCPDS No. 84-1108). The intensity of these peaks was small compared with the peaks associated with Ag, indicating a low amount of this material in the suspension [21,36,41].

### 3.4. Size Distribution of Nanoparticles Applying DLS

Figure 8 shows the results of this characterization for the three measurements performed on the same suspension of silver NPs. The highest intensity peaks appeared at 106.40 nm, 150.00 nm, and 148.40 nm. These values were used to determine the diameter in which the greatest number of NPs was concentrated. Although two of the measurements presented a similar maximum point, one of them exhibited a bimodal size distribution, grouping a small percentage of NPs in diameters of around 35.92 nm; on the other hand, the third measurement presented a shift in its maximum value towards smaller diameters that were greater than 100 nm. Another aspect of these measurements that should be highlighted is that the width of the peaks was considerable relative to the results obtained by other authors, which may have been due to differences in the synthesis process [42]. The aspects mentioned above led us to the conclusion that there was a high dispersion in terms of the size of the obtained AgNPs. This fact is attributable to the used synthesis method [40]. 

The results derived from the DLS analysis did not coincide with the information obtained from the UV–Vis spectrum, in which smaller sizes were recorded. According to Salguero Salas [23], the larger size of the AgNPs determined with DLS may have been caused by the fact that this technique allowed the dielectric layer to adhere to the surface of the NPs, and this influenced their Brownian movement in the suspension. Figure 9 shows a graphic representation of what the author expressed. However, this may not have been the only reason why this size variation occurred, so it was necessary to use another technique to confirm or rule out the presence of agglomerates in the suspension.

### 3.5. Morphological Analysis by SEM

Figure 10 shows a micrograph obtained with the scanning electron microscopy of the synthesized AgNPs. As can be seen, the AgNPs had an irregular shape, though with a tendency to adopt spherical geometries. In addition, the sizes were heterogeneous but remained in the range within which they could be considered nanomaterials, except for some agglomerates and/or aggregates that, according to [22], could have been formed due to variations in the suspension pH or, according to [6], because AgNPs have a natural tendency to form agglomerates or aggregates. It was concluded that the SEM results not only complemented the analyses carried out with other techniques but also cleared up doubts regarding the size of the synthesized AgNPs.

### 3.6. Microbiological Tests (Agar Diffusion)

Figure 11 is a photographic record of the growth of the fungi *Fusarium solani* and *Rhizopus stolonifer* cultivated in agar doped with concentrations of 0.1 mg/mL, 0.2 mg/mL, 0.5 mg/mL, and 1 mg/mL of AgNPs at observation times of 24, 48 and 120 h. Table 1 shows the growth diameters of the studied phytopathogens as a function of time and the concentrations of the NPs used to dope the agar. 

Figure 11 shows how doping the agar with silver AgNPs caused a delay in the growth of the fungi. For the fungus *Fusarium solani*, a growth retardation was recorded for concentrations of 0.1 to 0.5 mg/mL and a total inhibition was recorded for a concentration of 1 mg/mL. Similarly, this same situation occurred for the *Rhizopus stolonifer* fungus, though the behavior of *Rhizopus* was somewhat different; it was more aggressive and grew faster in both the standard sample and the treated samples.

The different behaviors of the two phytopathogens against AgNPs may have had several causes. For example, each one has a different way of growing and spreading in suitable media. The aggressiveness of the *Rhizopus* fungus and its greater resistance to the antifungal effect of AgNPs may be associated with the internal mechanisms of the fungus, which are related to the permeability of the cell membrane or the ability to develop exocytosis processes and eliminate a certain percentage of the silver NPs that enter the cell, as reported by the authors of [43] in their research on the anti-proliferative activity of silver NPs. 

Table 1 shows the growth measurements of each fungus, which were recorded as a function of time and AgNP concentration. These results demonstrate the aggressiveness presented by the phytopathogen *Rhizopus* in terms of its growth and greater resistance to NP treatment compared with *Fusarium*.

The set of results provided by the microbiological tests made it possible to verify that AgNPs had an antifungal effect on the studied phytopathogens. A schematic of the interaction of the AgNPs is presented in Figure 12. The antifungal effect was caused by different mechanisms by which NPs alter fungal cell viability. NPs tend to agglomerate around the cell membrane, generating reactive oxygen species that deactivate the cellular enzymes responsible for the processes of exchanging substances through the membrane and causing cell dehydration and the loss of nutrients [10,44]. In addition, previous studies have shown that another important mechanism related to the antifungal properties of these NPs is linked to the release of silver ions, since their cytotoxicity is diminished when brought into an alkaline pH environment [11,45]. This is consistent with the fact that most cells contain sulfur and phosphorus, which are weak bases that are part of ribonucleic acid and can be altered by the action of silver ions released by NPs, causing malfunctions and leading to cell apoptosis [46].

## 4. Conclusions

Using green chemistry and a coriander leaf extract as a reducing agent, silver nanoparticles were synthesized and their influence on the growth of the fungi *Fusarium solani* and *Rhizopus stolonifer* was evaluated in this study.

By means of UV–Vis spectroscopy, the obtainment of AgNPs was corroborated from the identification of absorbance peaks at wavelengths of around 428 nm, characteristic of these nanoparticles with diameters of between 40 and 60 nm.

The functional groups present in the extract of coriander (*Coriandrum sativum*), such as hydroxyls, carbonyls, carboxyls, amides, and phenols, were identified with Fourier Transform Infrared Spectroscopy (FTIR), which probably fulfilled the functions of reduction and stabilization in the process of nanoparticle synthesis.

Using the X-ray diffraction (XRD) technique and the Debye–Scherer equation, it was determined that the average crystallite size was 29.24 nm for the peaks associated with silver. Furthermore, it was established that the crystal structure of the material was FCC and that its lattice parameter corresponded to 0.40 nm. Likewise, the presence of silver oxide was detected.

The hydrodynamic size of the silver nanoparticles was determined using the dynamic light scattering (DLS) technique, which revealed a high particle size distribution with diameters greater than 100 nm, which (according to the literature) may be associated with the organic coating of nanoparticles or the formation of aggregates and agglomerates between them.

Scanning electron microscopy (SEM) was used to confirm the presence of nanoparticle agglomerates revealed by the DLS test, which may have been responsible for the increase in the observed size of the hydrodynamic diameter. The high size distribution of the obtained nanoparticles was also confirmed, as well as their irregular spherical shape, consistent with the obtained UV–Vis results.

The microbiological experiment by diffusion in agar showed that the silver nanoparticles had the ability to inhibit the growth of the fungi *Rhizopus stolonifer* and *Fusarium solani*. This inhibition was 100% in both fungi for a nanoparticle concentration of 1 mg/mL. In addition, the existence of a directly proportional relationship between the concentration and the percentage of inhibition was determined due to the physiological characteristics of the studied phytopathogens and the antifungal action mechanisms of the nanoparticles.

## Figures and Tables

**Figure 1 nanomaterials-13-00548-f001:**
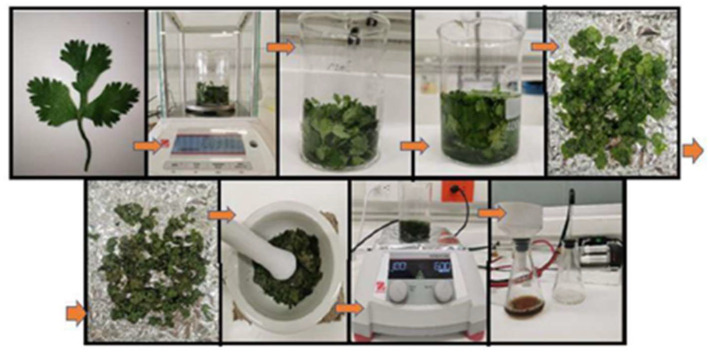
Coriander leaf extract preparation process.

**Figure 2 nanomaterials-13-00548-f002:**
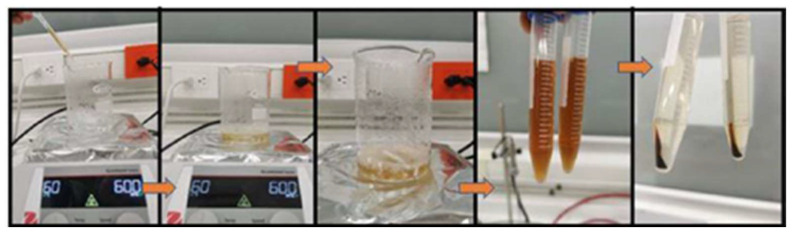
Reagent mixing process for the synthesis of AgNPs.

**Figure 3 nanomaterials-13-00548-f003:**
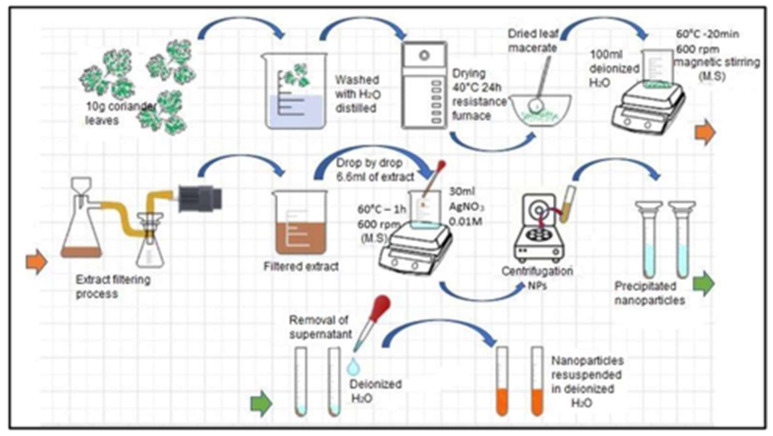
Precedence diagram illustrating the complete process of AgNP synthesis.

**Figure 4 nanomaterials-13-00548-f004:**
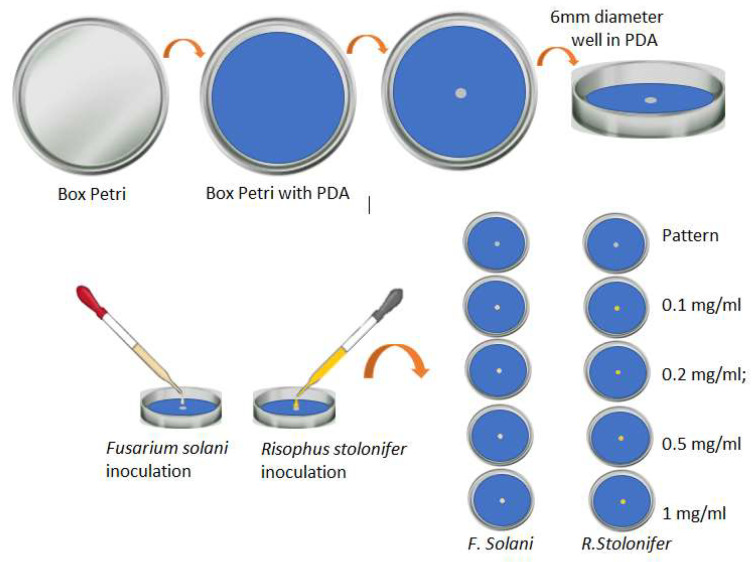
General scheme of the experiment carried out to evaluate the response of the two phytopathogens when placed in contact with AgNPs (own source).

**Figure 5 nanomaterials-13-00548-f005:**
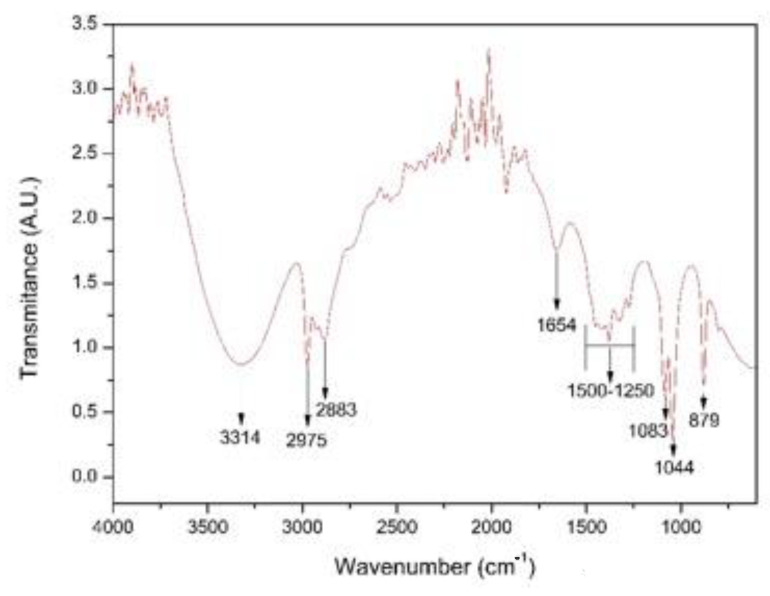
FTIR spectrum of the coriander extract (*Coriandrum sativum*).

**Figure 6 nanomaterials-13-00548-f006:**
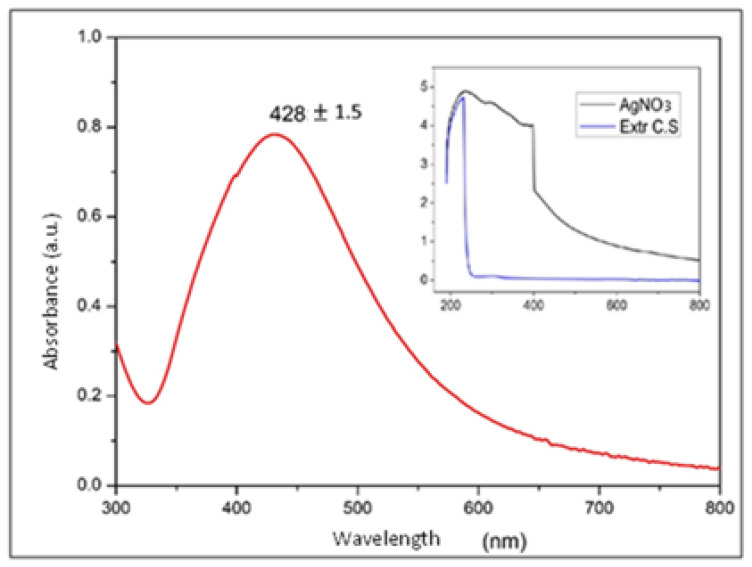
UV–Vis spectra at different scales of silver nanoparticles and the *Coriandrum sativum* extract with AgNO_3_.

**Figure 7 nanomaterials-13-00548-f007:**
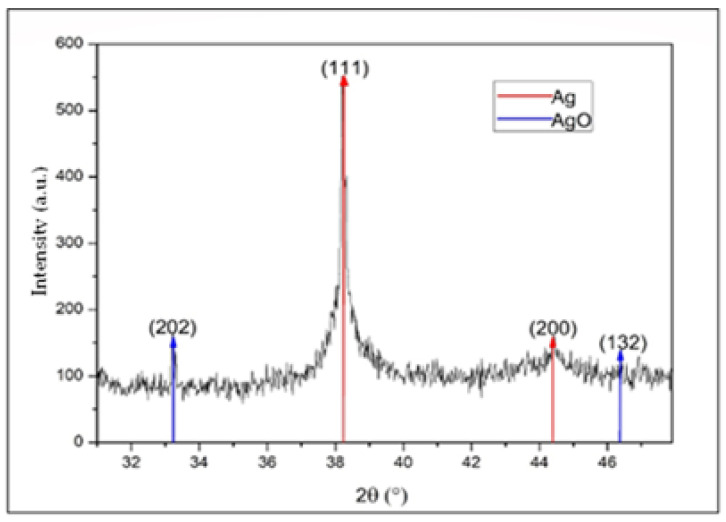
XRD diffractogram of silver nanoparticles between 2θ= 30° and 48°.

**Figure 8 nanomaterials-13-00548-f008:**
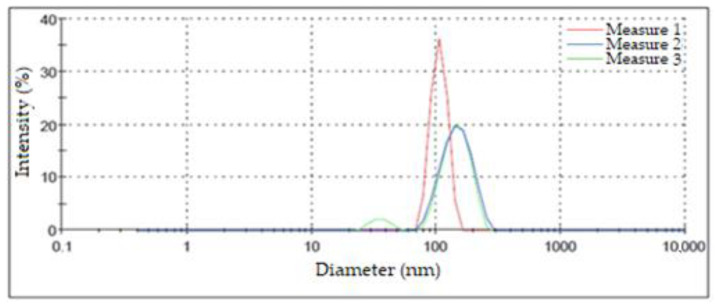
Particle size distribution according to DLS.

**Figure 9 nanomaterials-13-00548-f009:**
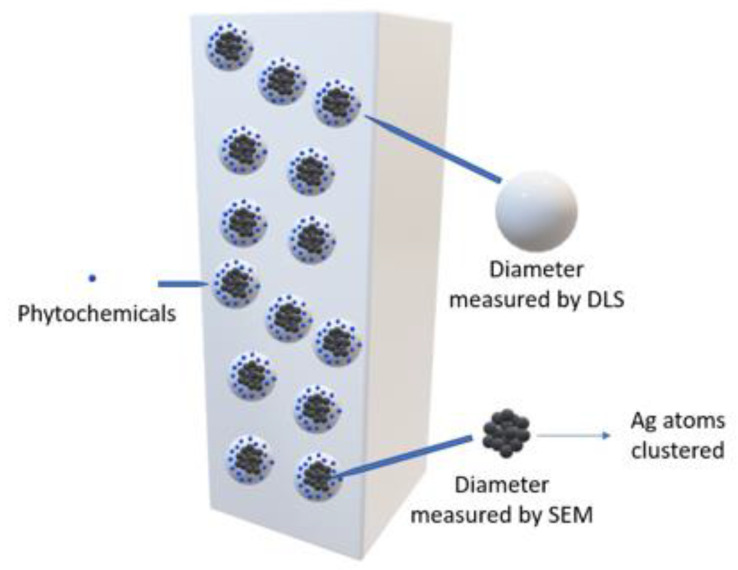
Diagram of the hydrodynamic diameter of AgNPs.

**Figure 10 nanomaterials-13-00548-f010:**
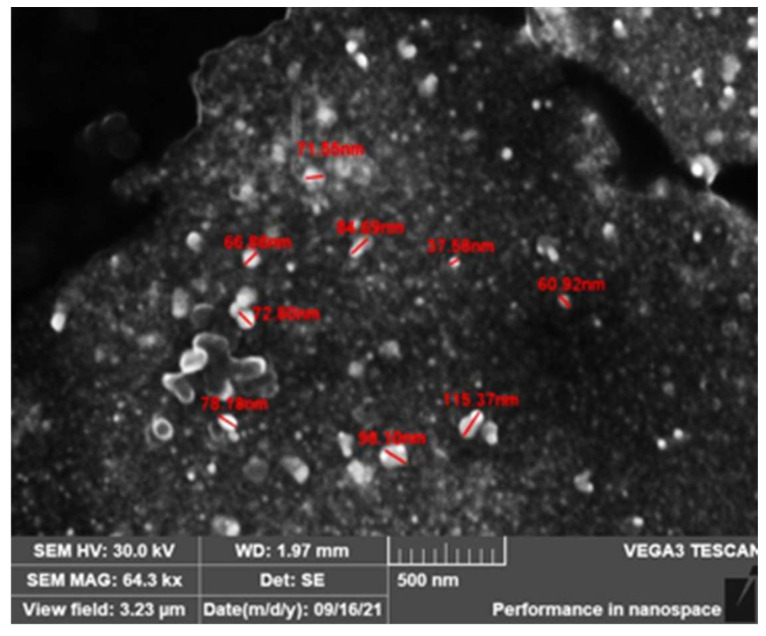
Scanning electron microscopy image of silver NPs.

**Figure 11 nanomaterials-13-00548-f011:**
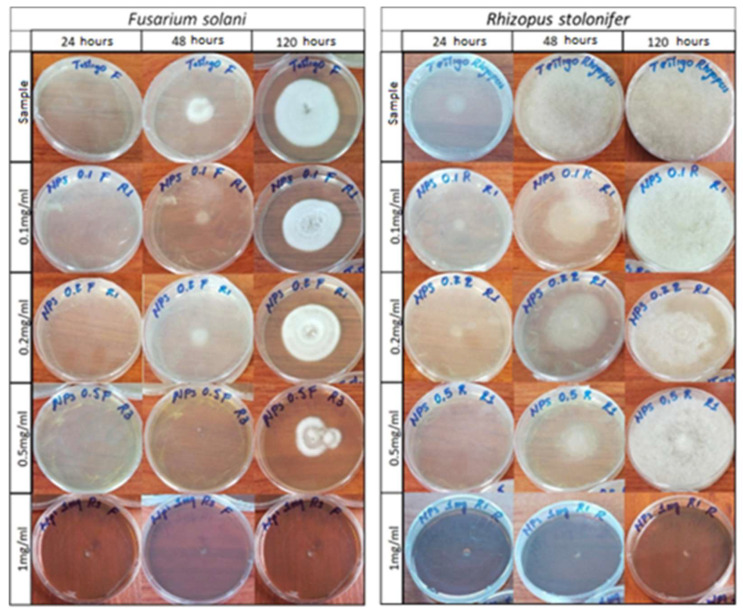
Photographic record of the growth of the fungi *Fusarium solani* and *Rhizopus stolonifer*.

**Figure 12 nanomaterials-13-00548-f012:**
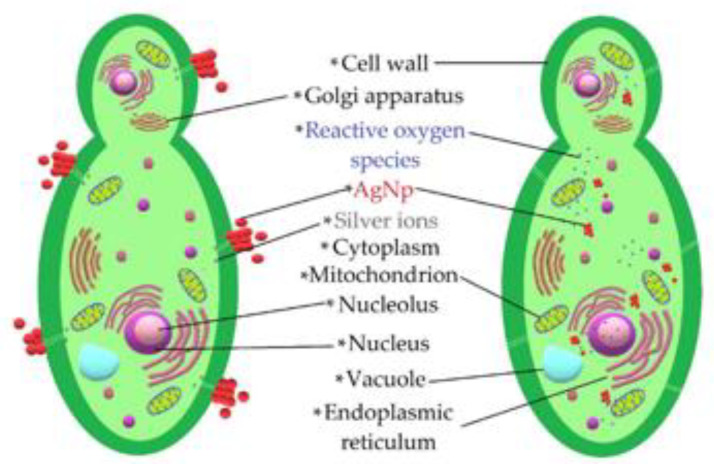
Schematic representation of the mechanisms employed by the AgNPs acting on phytopathogens.

**Table 1 nanomaterials-13-00548-t001:** Diameter values and standard deviation of the growth of *Rhizopus stolonifer* and *Fusarium solani* as a function of time for different concentrations of silver NPs.

	*Rhizopus stolonifer*			*Fusarium solani*		
Concentration (mg/mL)	Time					
	24 h	48 h	120 h	24 h	48 h	120 h
Testigo	15.30 mm	90.00 mm	90.00 mm	0 mm	24.11 mm	57.85 mm
0.1	9.24 ± 0.15 mm	40.49 ± 0.81 mm	90.00 mm	0 mm	12.09 ± 1.01 mm	45.94 ± 2.32 mm
0.2	10.01 ± 0.52 mm	45.37 ± 4.19 mm	90.00 mm	0 mm	12.65 ± 0.21 mm	47.19 ± 0.98 mm
0.5	0 mm	32.06 ± 1.85 mm	90.00 mm	0 mm	1.33 ± 0.58 mm	34.08 ± 2.04 mm
1	0 mm	0 mm	0 mm	0 mm	0 mm	0 mm

## Data Availability

No new data were created.

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
