# Peer review of "Evaluation of Antifungal Activity of Ag Nanoparticles Synthetized by Green Chemistry against Fusarium solani and Rhizopus stolonifera"

_nanomaterials, 2023, doi:10.3390/nano13030548_

Round 1
Reviewer 1 Report
Dear authors,
The author should provide some important results in the abstract section.
All the materials and their characteristics should provide materials and methods section.
Section 2.2. Preparation of the reducing agent should re-write the format of the journal.
2.4.1. Characterization of the extract; In order to determine the functional groups, present in the extract of coriander leaves and recognize components that make their contribution as a reducing agent and surfactant in the AgNPs synthesis process, a drop-shaped sample was placed on the Infrared Spectroscope cell. by Fourier Transformation (FTIR), Bruker ALPHA II. This equipment has a spectral resolution of 4 cm-1, and arrange of measurement between 350 - 8.000 cm-1. The spectra were automatically analyzed by the software belonging to the equipment.
The author should re-write the procedure in meaningful. It should be explaining such as Infrared Spectroscope cell, measurement between 350 - 8.000 cm-1, and the spectra were automatically analyzed by the software belonging to the equipment.
This spectrum is evidence that the measured nanoparticles produce a strong absorption band by the surface plasmon in the visible region, centered at wavelengths between 420 and 440 nm, a typical range for AgNPs with sizes between 40 and 60 nm in diameter. The author should xpalin the aforementioned statements. How is calculated the typical range for AgNPs with sizes between 40 and 60 nm in diameter from UV-Vis measurements?
UV-Vis spectrums are not clear. The author should study again.
The author should provide TEM analysis of AgNPs and their size distribution analysis.
Conclusion should be revised to show the outstanding point of this work.
Author Response
Regarding figures 9 and 12, I have the following observations:
1. Figure 9: This figure was changed as it was not possible to contact the author of the figure. For this reason, another scheme different from the one that had been initially put in, but which explains the phenomenon, was made.
2. Figure 12: The reference of the figure, which is free copyright, was included.

Reviewer 2 Report
The authors synthesized and characterized Ag nanoparticles with natural reductants from coriander extraction and use control experiments showing the antifungal properties of these synthetic nanoparticles. The green synthesis method of Ag nanoparticles from different leaves extraction is not a novel topic so authors need to provide good characterization to prove Ag nanoparticles in this work have some unique or better properties than other work.
Also, the characterization methods in this paper contain some fundamental defects and mistakes.
Line 132, the range “350-8.000 cm-1” is confusing, an ordinary commercial FTIR ranges from 4000 to 400 cm-1. Please claim this and provide the name and version of the FTIR software used in this paper.
Line 142, provides electron voltage used in characterization rather than the magnification range of SEM.
Line 145 “in 2theta”?
Line 147, in the silver nanoparticle concentration test, do nanoparticles solution directly injected into AAS? Did you dissolve silver nanoparticles into ions before the test? Did you use any calibration standard to determine the concentration?
Line 179, board peak around 3400 cm-1 is O-H from water because described in your experimental section, extraction is tested in an aqueous solution. It is important to keep the sample away from humidity for FTIR because the water peak can shield lots of signals.
Line 186 why there are halogen bonds in the extraction? Do you have any estimation on functional molecules from extraction?
Some problems need to be emphasized in size determination.
Size, shape, coating, and oxidation can cause the blue- or red-shift of silver nanoparticle's UV spectra. It is not convinced to determine the size by the center of the absorption peak.
Line 210, how can you determine error is 1.5 cm-1 in wavelength?
Figure 7, XRD, why does the graph only show a range from 30-48? When showing the XRD diffraction pattern, at least the first three strongest peaks should be presented. Also, labels of Ag/AgO in the graph are inverted.
DLS, please add labels for three peaks. Why three measurements of the same sample are not consistent?
Figure 10, the AgNPs are too concentrated and aggregate together. To get a better view, you need to dilute the aqueous solution to an appropriate concentration and then take SEM photos.
Table 1 Please claim how many culture plates were processed in each group.
The authors reviewed two possible toxicities mechanisms from published literature papers but lack the analysis based on their work. For example, in Line 326, the authors mentioned that Ag ion toxicity is pH related, but didn’t provide the pH value in your fungi culture plate.
In conclusion, this paper didn’t bring novel things to the AgNPs antimicrobials study. I recommend this paper be revised carefully in characterization methods and significant discussion.
Author Response

(The authors gave the same response as above.)

Round 2
Reviewer 1 Report
Revised version is suitable for publication.
Reviewer 2 Report
The authors add detailed discussion based on all comments. I recommend the paper be accepted and published.